# Ophthalmomyiasis Externa and Importance of Risk Factors, Clinical Manifestations, and Diagnosis: Review of the Medical Literature

**DOI:** 10.3390/diseases11040180

**Published:** 2023-12-11

**Authors:** Hugo Martinez-Rojano, Herón Huerta, Reyna Sámano, Gabriela Chico-Barba, Jennifer Mier-Cabrera, Estibeyesbo Said Plascencia-Nieto

**Affiliations:** 1Sección de Posgrado e Investigación, Escuela Superior de Medicina del Instituto Politécnico Nacional, Plan de San Luis y Díaz Mirón s/n, Colonia Casco de Santo Tomas, Delegación Miguel Hidalgo, Ciudad de México 11340, Mexico; saidpn@yahoo.com.mx; 2Coordinación de Medicina Laboral, Instituto de Diagnóstico y Referencia Epidemiológicos, Francisco de P. Miranda No. 177, Colonia Unidad Lomas de Plateros, Ciudad de México 01480, Mexico; 3Laboratorio de Entomología, Instituto de Diagnóstico y Referencia Epidemiológicos, Francisco de P. Miranda No. 177, Colonia Unidad Lomas de Plateros, Ciudad de México 01480, Mexico; cerato_2000@yahoo.com; 4Departamento de Nutrición y Bioprogramación, Instituto Nacional de Perinatología, Secretaría de Salud Montes Urales 800, Lomas de Virreyes, Alcaldía Miguel Hidalgo, Ciudad de México 11000, Mexico; ssmr0119@yahoo.com.mx (R.S.); gabyc3@gmail.com (G.C.-B.); jennifer.mier@gmail.com (J.M.-C.)

**Keywords:** external ophthalmomyiasis, *Oestrus ovis*, *Dermatobia hominis*, conjunctival myiasis, external ocular myiasis, global warming, red eye, diagnosis

## Abstract

Objective: This review aims to describe and compare the risk factors, clinical course, diagnosis, and parasitologic features of external ophthalmomyiasis. We also discuss the different preventive measures and the effect of global warming in a large case series reported from January 2000 to December 2022. Methods: We perform a literature review of reported cases of external ophthalmomyiasis to determine the clinical characteristics, therapeutic management, and information on the organisms that most commonly cause external ophthalmomyiasis. Results: A total of 312 cases of external ophthalmomyiasis were recorded. The most common causative organism was *Oestrus ovis* (Linnaeus) (Diptera: Oestridae), accounting for 72.1% of cases, followed by *Dermatobia hominis* (Linnaeus Jr. in Pallas, 1781) (Diptera: Oestridae) at 5.4%, *Lucilia sericata* (Meigen) (Diptera: Calliphoridae) at 0.96%, and *Chrysomyia bezziana* (Villeneuve) (Diptera: Calliphoridae) at 0.96%. Before experiencing symptoms, 43.6% of cases reported having direct contact with flies or being hit in the eye, 33% reported no associated risk factors, 9.3% reported living with sheep and goats, and 7.7% reported a history of foreign objects entering the eye. The most common symptoms for those affected by *O. ovis* were sudden onset, including a foreign body sensation and movement, redness, tearing, itching, swelling, irritation, photophobia, burning, and ocular secretion. In cases caused by *D. hominis*, symptoms included eyelid edema, pain, redness, itching, movement within the lesion, ocular irritation, and discharge. Regarding occupational or recreational activity, agriculture, and livestock had the highest incidence, followed by trades and technical activities, being a student, and having traveled to an endemic region for work or recreation. Conclusion: Patients with red eyes may suffer from external ophthalmomyiasis, which requires a thorough examination to diagnose and treat it early. Moreover, as the temperature increases due to climate change, it is essential to consider how this will affect the spread of different pathogens.

## 1. Introduction

Myiasis is a term that defines a condition characterized by the infestation of a mammal by fly larvae to feed on its tissues [1,2]. The condition does not result from adult flies. Myiasis can affect humans and animals, leading to various clinical presentations [3,4,5]. Myiasis is a type of infestation categorized as obligate, facultative, or accidental. Obligate myiasis is mainly caused by fly larvae (*Dermatobia*, *Cordylobia*, *Wohlfahrtia*, and *Oestrus*), which cannot obtain enough nutrients from their mother fly. In animals, blowfly larvae are the primary cause of myiasis and can also affect humans as a zoonosis. The most common type of myiasis among humans is caused by *D. hominis* [6,7], whose larvae are spread via mosquitoes by laying their eggs on mosquitos’ wings. *Cordylobia anthropophaga* (Blanchard 1871) (Diptera: Calliphoridae) is spread through urine and feces. Facultative myiasis is caused by bluebottle flies (*Phormia*, *Lucilia*, and *Musca*), and some of their larvae (*L. sericata*) release bacteriolytic enzymes that can be used to sterilize infected wounds [8,9].

Myiasis is becoming more common in environments with poor hygiene [3], primarily where human activity occurs. Although rare in humans, it is observed in tropical and subtropical areas with suboptimal housing conditions [10]. Poor hygiene, alcoholism, trauma, senility, mental or neurological diseases, immunosuppression, diabetes, malnutrition, and suppurative lesions [11] are all risk factors associated with human myiasis. The prevalence of myiasis depends on climatic and ecological factors as well as the fly population and susceptible animals [12]. Larvae can invade various body parts, including the mouth, nose, eyes, lungs, ears, sinuses, anus, brain, and vagina [10]. Ophthalmomyiasis, or ocular myiasis, is a form of myiasis that specifically affects the human eye. However, it represents less than 5% of all human myiasis cases [5]. Ophthalmomyiasis can range in severity from mild irritation to blindness or even death [13]. Depending on the location of the infestation, myiasis is classified as external, internal, or orbital. External myiasis occurs when larvae are present in the conjunctiva, sclera, eyelids, and tear ducts, including palpebral and conjunctival myiasis. Internal myiasis occurs when larvae penetrate the eye. Meanwhile, orbital myiasis is the least common type and occurs when larvae invade the orbit, potentially leading to the rapid destruction of the eyeball [3,5,13,14].

In 1900, Keyt reported the first case of ophthalmomyiasis [15]. The most common cause of ophthalmomyiasis externa is *O. ovis* (“sheep fly”) larvae [16]. Dipterous flies known to cause ophthalmomyiasis externa include the sheep nose fly (*O. ovis*) and the human bot fly (*D. hominis*), and are endemic to tropical countries, with other examples including rodent or rabbit bot flies of the genus *Cuterebra* (Diptera: Oestridae) and equine bots of the genus *Gasterophilus* (Diptera: Gasterophilidae), the housefly (*M. domestica*), and the bovine fly (*Hypoderma*). Of these, *O. ovis*, which is an obligate parasite of sheep and goats, with humans occasionally becoming accidental hosts, has a cosmopolitan distribution and has been reported to be the most common causative agent of ophthalmomyiasis externa [17,18].

External ophthalmomyiasis is rare in Mexico, with only two cases reported so far [19,20]. Our comprehensive literature review provides valuable information on the various risk factors, clinical course, diagnosis, parasitological and environmental characteristics, and treatment options for this condition. In addition, we discuss preventive measures and the effect of global warming based on a case series reported between January 2000 and December 2022.

## 2. Methods

We conducted a thorough search of electronic databases (Latin American and Caribbean Literature in Health Sciences, http://lilacs.bvsalud.org; Scientific Electronic Library Online, http://www.scielo.org; PubMed, http://www.ncbi.nlm.nih.gov/pubmed (accessed on 1 June 2023); EBSCOhost, http://www.ebscohost.com; and Google Scholar, http://scholar.google.com) and individual journals, including articles in English and other languages (German, Turkish, French, and Italian) published between 1 January 2000, and 31 December 2022, on external ophthalmomyiasis, external ocular myiasis, and conjunctival myiasis. Our research strategy was comprehensive, utilizing a combination of general terms (“ophthalmomyiasis”, “external ophthalmomyiasis”, “superficial ophthalmomyiasis”, “human ophthalmomyiasis”, “conjunctival myiasis”, “ocular myiasis”, “external ocular myiasis”, “palpebral myiasis”, “accidental myiasis”, “oestrosis”, “*Oestrus ovis*”, “*Dermatobia hominis*”, “*Cochliomyia hominivorax*”, “*Chrysomyia bezziana*”, “*Lucilia sericata*”, “*Musca domestica*”, “myiasis”, “sheep fly”, “sheep blowfly”, “diptera larvae”, “maggots”, “Ivermectin”, “sheep”, “zoonosis”, “acute presentation”, “conjunctivitis/Differential diagnosis”, “unilateral conjunctivitis”, “acute unilateral conjunctivitis”, “red eye”, “keratouveitis”, “keratitis”, “ocular”, “ocular infestation”, “ocular foreign body”, “red eye”, and their combinations) and cross-referencing to identify relevant articles. We only selected articles that reported external ophthalmomyiasis and met the eligibility criteria.

## 3. Results

### 3.1. Epidemiological Information

Myiasis is a condition usually seen in rural or low-income regions where people are in close proximity to farm animals [21,22]. Nevertheless, there has been an increase in reported cases in urban areas without known risk factors [23,24,25]. The yearly occurrence rate in Africa and the Middle East is 10 cases per 10,000 inhabitants, but instances have also been recorded in Australia, North America, and Southern Europe [26].

After searching various databases, we found 117 articles that discussed 312 cases of external ophthalmomyiasis from 2000 to 2022. Among these cases, 208 were male, 102 were female, and 2 did not disclose their sex, resulting in a male-to-female ratio of 2:1. The average age of the patients was 32.1 years, with the youngest case being a few days old, and the oldest being 91 years old. It is worth noting that this condition predominantly affects adults aged 20–59 and adolescents aged 10–19.

India reported the highest incidence of cases from 2000 to 2022, accounting for 62 (19%), followed by Jordan with 50 (16.03%), Turkey with 45, Iran with 27, and Libya with 22. Notably, 17 countries reported one case each, with the patient in each case having traveled to endemic countries for tourism or work.

*Oestrus ovis* larvae were the culprit in 225 cases, representing 72.12% of all reported cases. In 56 instances, the cause remained unidentified. *D. hominis* was pinpointed in 17 cases, while *L. sericata* and *C. bezziana* were each discovered in 3 cases (Table 1). Agriculture and livestock were the most common activities associated with the reported cases, followed by trades/technical activities and recreational or work trips to endemic countries.

### 3.2. Clinical Features

Among the patients, 43.6% reported a sensation of insects buzzing around their faces or being hit in the eye before the onset of symptoms. In 33% of cases, no identifiable risk factors were associated with the symptoms. It was observed that approximately 9.3% of cases had a history of coexisting with sheep and goats, while 7.7% reported a previous incident of foreign objects entering the eye.

External ophthalmomyiasis caused by *O. ovis* larvae presents with a sudden onset of symptoms such as foreign body sensation, redness, tearing, itching, swelling, irritation, photophobia, burning, and ocular discharge. Bilateral involvement was reported in four cases [27,28]. The slit-lamp examinations revealed highly mobile tiny larvae (1–2 mm long) fleeing from light, conjunctival congestion, hyperemia, chemosis, palpebral edema and hyperemia, epiphora, mucoid discharge, subconjunctival hemorrhage, punctate keratitis, epithelial erosion, corneal abrasion, and palpebral cellulitis. The average number of larvae observed was 7.2 (1–30). *D. hominis*-caused external ocular myiasis presents with symptoms such as edema of the upper or lower eyelid, edema of the inner or outer canthus of the eye, pain, redness, itching, a sensation of movement inside the lesion, ocular irritation, and serosanguinous discharge. The ophthalmologic examinations revealed edema and palpebral hyperemia, lesions with the fistulous tract, serosanguinous secretion, exudate, and movement of part of the worm at the end of the fistulous tract. Only one larva was observed in most cases, while one case reported three larvae [17].

*Lucilia sericata* can cause external ophthalmomyiasis, leading to symptoms such as foreign body sensation, pain, ocular discharge, itching, tearing, and redness. During the ophthalmologic examinations, tiny larvae, hyperemic and congestive conjunctiva, chemosis, serous discharge, palpebral hyperemia, and periorbital edema were commonly observed. On average, 12.3 larvae were found (ranging from 2 to 19 per eye). Similarly, external ophthalmomyiasis caused by *C. bezziana* may result in symptoms such as worms emerging through the lesion, edema, fetid secretion, itching, and redness. The ophthalmologic examinations revealed periorbital lesions, worms, conjunctival hyperemia, congestion, and periorbital edema. On average, 16.7 worms were recovered during the examinations (ranging from 1 to 26 per eye). The general characteristics and clinical pictures of external ophthalmomyiasis cases reported between 2000 and 2022 are shown in Appendix A.

### 3.3. Diagnosis

A high index of suspicion is necessary for both the ophthalmologist and the entomologist to make a correct diagnosis [29]. The symptoms of external ophthalmomyiasis are non-specific. Therefore, it can be misdiagnosed as any other type of acute viral or bacterial conjunctivitis if myiasis is not considered by clinicians [30]. In the case of *O. ovis* ophthalmomyiasis, the manifestations of human conjunctival myiasis are usually short-lived and self-limiting, as the larvae cannot develop further and die within 10 days [30].

In the vast majority of cases, ophthalmomyiasis diagnosis is made via slit-lamp biomicroscopy, but in remote endemic regions, the dermatoscope has been successfully used as a portable tool for the diagnosis of ophthalmomyiasis externa [31,32].

During a slit-lamp examination, *O. ovis* shows negative phototaxis, with the small, translucent larvae moving away from the light quite quickly. This could lead to misdiagnosis if the larvae are missed, as it is possible for them to hide in the fornix of the patient’s eye. The morphological properties of *O. ovis* larvae are relatively typical but differ in some cases, so molecular verification of the evidence is recommended [33]. However, this is not a routine procedure.

In most cases covered in previous studies, exact taxonomic classification of the extracted larvae is performed, providing an idea of the potential risk of intraocular complications. In addition, in previous studies, larvae were preserved in 70% alcohol and sent to a specialist, who performed the identification and taxonomic classification [34].

Differential diagnoses include follicular conjunctivitis, foreign body conjunctivitis, catarrhal conjunctivitis, and other infestations, such as those of tiny *Paederus* spp. (Coleoptera: Staphylinidae) beetles, which sometimes hit the conjunctiva by accident, releasing poison and causing irritation and subsequent ophthalmic problems [35]. The significant criteria for the diagnosis of ophthalmomyiasis are as follows: the sudden sensation of a foreign body causing sudden itching and tearing occurring while the patient is in an endemic area during warmer months, with the subject not necessarily noticing any flies [36].

### 3.4. Treatment

The treatment for ophthalmomyiasis involves removing the larvae with tweezers, followed by anti-inflammatory drugs and local antibiotics [37]. This method provides immediate relief for patients without adverse effects on their vision. For *O. ovis* infestations, manual extraction, and local antibiotic application were used in 109 cases (52.8%), while manual extraction, antibiotic application, and local steroid use were employed in 86 (41.3%). For *D. hominis* infestations, surgical removal was performed in 10 cases (58.8%), and surgical removal with antibiotic administration was carried out in 4 cases (23.5%). *L. sericata* infestation was treated with manual removal and antibiotic application in two cases and with steroid application in only one case. In cases of external ophthalmomyiasis caused by *C. bezziana*, manual larva extraction, and local antibiotic application were used in two cases. Manual extraction with systemic application of antibiotics and ivermectin was reported in only one case (Table 2).

### 3.5. Complications

External ophthalmomyiasis caused by *O. ovis* is generally associated with a low incidence of complications, such as subconjunctival hemorrhage or blindness [19,38]. Nonetheless, a few rare cases of severe complications have been documented, including diffuse stromal keratitis and anterior uveitis [39]. In Korea, 18 cases of *O. ovis* infestation were reported [40], with corneal infiltration occurring in 3 cases. Although infestation generally starts with first-instars, third-instars have been discovered in patients with severe illnesses such as cancer or immunodeficiency [41].

## 4. Discussion

External ophthalmomyiasis caused by Diptera insect larvae in the conjunctival sac is the most common form of ophthalmomyiasis. Our review (including 117 publications reporting 312 cases) shows that it is prevalent in regions where sheep or goats are raised, including India [1], Jordan [18], Turkey [37], Iran [40], Libya [42], Tunisia [43], Spain [44], and Italy [45]. In the past decade, there have been reported cases of this infection in countries where autochthonous cases were previously unknown, such as the Czech Republic [46], Bulgaria [47], Australia [48], China [49], Serbia [19], Jamaica [50], South Africa [51], Morocco [52], Japan [53], Indonesia [54], Canada [55], Barbados [56], Belgium [57], Iraq [58], United Kingdom [59], Honduras [60], and Bolivia [61]. Despite its global prevalence [10,18], most reported cases are still concentrated in India [2,27,29,37,62,63,64,65,66,67], Turkey [25,28,43,68,69,70,71,72,73,74], Iran [20,40,75], and Libya [18,76].

India has experienced a substantial number of external ophthalmomyiasis cases over the last two decades, as evidenced by nine case series studies, with Misra et al. conducting the largest study in 2008, including 13 cases [37]. Other studies by Singh et al. in 2012, Sucilathangam et al., and Choudhary et al. in 2013 reported 3, 10, and 7 cases, respectively [27,38,77]. Singh et al. reported three cases in 2002 [78], while in the other 22 reports, one or two cases were reported [2,13,20,29,30,62,63,64,65,79,80,81,82,83,84,85,86,87]. The causative agents identified in India were *O. ovis*, *Musca* ssp., *Musca domestica* Linnaeus (Insecta: Diptera: Muscidae), and *C. bezziana*. Given the socioeconomic conditions and climate change, it is imperative to address ophthalmomyiasis as a potential health concern in India.

According to Thabit et al. [88], 49 cases of external ophthalmomyiasis were found in Jordan, but the causes were not determined. It was observed that most of the affected individuals were male, with an average age of 31.6 years, which is consistent with the previous research comprising 312 cases. Alameri et al. [42] reported only one case caused by *O. ovis* in 2014. However, the actual number of cases may be higher than reported due to underreporting and the mild symptoms experienced in most cases.

Turkey has been experiencing a surge in cases of external ophthalmomyiasis over the past 22 years, which cannot be attributed to known risk factors. Turkey ranks third in the world in the number of reported cases. Various studies have reported case numbers ranging from one to twelve [25,26,29,43,68,69,70,71,72,73,89,90,91,92], with the causative agent predominantly being *O. ovis* and *C. bezziana* in only one case [89]. Interestingly, cases have occurred more frequently in urban areas without known animal contact.

### 4.1. Risk Factors

External ophthalmomyiasis cases have been reported across Asia, Europe, and South America from 2000 to 2022, with a high incidence in countries such as India [2,27,29,37,62,63,64,65,66,67,79], Jordan [42,88], Turkey [25,29,43,68,69,70,71,72,73,74], Iran [20,40,75], and Brazil [17,19,93,94]. As shown in Table 1, the geographical distribution of the flies responsible for this disease is mainly in warm and humid tropical or subtropical areas ideal for fly breeding [10]. External ophthalmomyiasis is more prevalent in pastoral or rural areas than urban centers, especially in developing countries with inadequate basic sanitation and garbage disposal [37,43,88]. Nevertheless, due to the rise of international travel, infection can also occur in non-endemic areas [46,49,53]. Since larvae enter the eye outdoors, men are more likely to be affected than women, as demonstrated in our review, with a higher proportion of cases occurring in males (with a 2:1 male-to-female ratio) [78].

Unfortunately, occupations such as farming, shepherding, and veterinary work come with inherent risks. Studies conducted in India have revealed that approximately 51.75% of individuals with these jobs are at an increased risk of infestation [37,38,77]. Research by Hira et al. [95] and Masoodi et al. [75] found a higher incidence of ophthalmomyiasis in areas where sheep and goats are raised, indicating that myiasis is an occupational disease for farmers and shepherds. However, in this review, only 32.17% of cases were associated with occupational activities. Ophthalmomyiasis is common in urban areas, affecting teachers, office workers, nurses, and students [36,96,97].

It is important to note that tourist and recreational activities accounted for 10.49% of all reported cases. Tamponi et al. [98] documented three instances of external ophthalmomyiasis in Italian tourists, while Cucera et al. [99] reported two cases in German tourists. Dorchies [100] noted that external ophthalmomyiasis is usually accidental.

Various factors can heighten the likelihood of developing ophthalmomyiasis [89,101,102]. These include eye infections or injuries (whether from surgery or trauma), advanced age, persistent illnesses such as infections or cancer, HIV infection, alcoholism, diabetes, psychiatric disorders, poor overall health, homelessness, and close contact with farm animals.

The use of manure from livestock grazing pens as a fertilizer in urban areas can be a source of infestation in areas that do not have livestock because the manure from host animals is used as a fertilizer in gardens or sports fields, bringing unhatched *O. ovis* pupae closer to human hosts [96]. This exposure is a controllable risk factor for external ophthalmomyiasis, and it is recommended to avoid the use of manure in gardens or recreational areas frequented by people to prevent the spread of infestations [103]. Manure from grazing pens may be the only source of infestation in areas without livestock.

### 4.2. Etiological Agent

In 2020, Pupić-Bakrač et al. [104] documented two cases of external ophthalmomyiasis caused by *O. ovis* in an urban area of Croatia. Another of their studies revealed 260 cases of human ophthalmomyiasis, with 259 (99.62%) cases of external ophthalmomyiasis (*O. ovis*) and only 1 case of internal ophthalmomyiasis (0.38%). Similarly, Akbarzadeh et al. [105] and Abdellatif et al. [18] also identified *O. ovis* as the cause. Our review of cases across a wide geographical distribution [27] Asia [37,53,54,58,65,75,88,102,106,107], Europe [19,31,36,45,57,59,99,108,109,110], Africa [18,51,52,76], Australia [48], and America [17,19,55,56,93,111,112,113,114,115] published in the last 22 years found that 225/352 (72.12%) cases had external ophthalmomyiasis (*O. ovis*), with 18% of cases not reporting the causative agent [27]. This differs from studies that identified *O. ovis* as the cause in endemic Mediterranean countries [18,104,105], including cases reported since the 1970s. Most patients engaged in agricultural activities (such as sheep/goat breeders, ranchers, farmers, and peasants) are prone to ophthalmomyiasis (*O. ovis*), as reported by Pupić-Bakrač [104], Thakur [3], and Abdellatif [18]. However, our findings contrast theirs, as only 38.4% of cases of external ophthalmomyiasis were associated with risky work activities. This behavior is mainly attributed to the effect of global warming, creating habitats more suitable for the survival of O. ovis and not to an increase in reports [38,116,117,118].

Ophthalmomyiasis cases reported in Turkey [76] and Iran [105,119] were primarily caused by *O. ovis*, with additional cases attributed to various Diptera such as *Sarcophaga* sp. (Diptera: Sarcophagidae), *Rhinoestrus purpureus* (Diptera: Oestridae), *Hypoderma bovis* (Diptera: Oestridae), and *C. bezziana*. Similar cases have been reported in Middle Eastern and Mediterranean countries over the past 22 years. *O ovis* is the most common cause, but occasionally, other Diptera of different genera, such as *C. bezziana* or *Sarcophaga* sp., are the cause. It is crucial to address this issue promptly to prevent further spread and harm.

In the case of the Americas, the majority of reported cases of external ophthalmomyiasis were caused by fly larvae from South America (*D. hominis*) [55] or Africa (*C. anthropophaga*) [119], and these cases generally occurred in tourists returning from endemic regions [120]. For North American patients without a recent travel history, it represented a diagnostic challenge [111]. *Cuterebra* species were the most common causative agent of myiasis in patients residing in North America who had no history of travel to endemic regions [121]. *Cuterebra* species are large bee-like flies that are endemic to the northeastern United States and southeastern Canada [111].

### 4.3. Clinical Manifestations

Clinicians must consider ophthalmomyiasis a possibility in patients exhibiting non-specific symptoms, as demonstrated in this study and various others [26]. Failure to do so in endemic areas or tourists returning from such regions may lead to mistaking ophthalmomyiasis for other forms of conjunctivitis [30]. Symptoms typically resemble acute catarrhal conjunctivitis and include burning, stinging, itching, increased tearing, and the sensation of a foreign object in the eye [10].

Regarding ophthalmomyiasis caused by *O. ovis*, patients commonly report foreign body sensations, red eyes, and excessive tearing, all of which indicate significant irritation in the affected eye [38,78,104]. These symptoms are consistent with the behavior and structure of the larvae, which are equipped with hooks and oral spines that can irritate as they move across the conjunctiva and cornea [78]. Meanwhile, cases of *D. hominis* commonly result in symptoms such as eyelid swelling, redness around the eye, and a feeling of movement in the lesion, which are consistent with other reports [17,122]. External ophthalmomyiasis induced by *L. sericata* typically causes pain, foreign body sensation, itching, and excessive tearing, which is similar to the findings of another study [20]. Finally, *C. bezziana* often results in symptoms such as the emergence of maggots from the wound, smelly discharge, swelling, and itching, which have been reported by other researchers [10,123].

Research findings support that external ophthalmomyiasis symptoms can manifest immediately or several hours after contact with the fly, depending on whether the fly’s eggs or larvae are active. The onset of symptoms is directly related to the time it takes for the eggs to hatch. It is important to note that clinical signs appear immediately after contact with *O. ovis*, while with *M. domestica* or *D. hominis*, symptoms may appear a few hours later. These findings are supported by extensive research [6,10,78,124,125].

*Oestrus ovis* Infestations may cause brief and self-limiting ophthalmomyiasis symptoms, as the larvae cannot survive for more than ten days [64]. However, invasive manifestations may occur in rare cases depending on host factors [104,126]. Internal ophthalmomyiasis has been reported [127], but it is uncommon. External ophthalmomyiasis usually affects only one eye, but there have been more frequent cases of bilateral involvement [2,51] in recent years, with ten reported cases [14,19,27,28,37,86,128,129,130] over the past 22 years. *O. ovis*, *S. argyrostoma*, *M. domestica*, and one unreported species were responsible for the infestations, demonstrating that bilateral involvement is still uncommon but not rare.

Over 22 years, numerous case reports strongly indicate the possibility of experiencing a foreign object sensation in the eye after being struck by a fly [3]. It is essential to accurately differentiate this particular sensation from other conditions that may present similar symptoms [3]. In humans, the larvae are usually found in the outer part of the eye, causing discomfort and irritation [3]. The larvae use their mouth hooks to attach to the eye, contributing to this discomfort [3].

External ophthalmomyiasis is characterized by severe local conjunctival inflammation that can cause uncomfortable symptoms such as foreign body sensation, irritation, redness, photophobia, lacrimation, and decreased visual acuity. Signs of this condition include erythema, periorbital edema, conjunctival edema, hemorrhages, chemosis, and superficial punctate keratitis. Additionally, some patients may experience a sensation of movement within the eye [2,51]. In some cases, the eyelids may also be affected, leading to edema [120], which can be mistaken for a chalazion or preseptal cellulitis. However, identifying the respiratory pore and feeling contortion within the lesion can help diagnose cases caused by *D. hominis* [130]. It is imperative to promptly seek medical attention in case of experiencing any of these symptoms.

### 4.4. Diagnosis

With over 85,000 species in the Diptera order, only a select few are responsible for causing ophthalmomyiasis [131]. Expert identification of the larvae causing external ophthalmomyiasis is based on the classification of their posterior spiracle and a cephaloskeleton structure [132]. It is of utmost importance to identify the species of flies involved, as some can cause substantial harm to deeper eye tissues and affect vision [3]. While viral and bacterial infections are the most common causes of red eyes, parasitic infestations can also lead to unilateral red eyes [30]. Therefore, external ophthalmomyiasis should be considered a possible cause of pink eye, even in healthy individuals residing in large cities [73], as approximately 15% of cases reported over the last two decades have occurred in healthy people with no history and living in large cities [47,49,96].

External ophthalmomyiasis is a condition that can be diagnosed by observing larvae or worms in the conjunctiva and/or adnexa. It is essential to have a high index of suspicion to correctly diagnose the condition, as small numbers of larvae may be present, which can be mistaken for acute conjunctivitis [133]. Our findings are consistent with an average of seven larvae reported in patients from Libya [18] who had ophthalmomyiasis caused by *O. ovis*. In cases caused by *D. hominis*, only one larva was reported in each, except in a two-month-old infant with one larva in the left upper eyelid and two in the left lower eyelid [17]. For external ophthalmomyiasis caused by *L. sericata* and *C. bezziana*, the mean numbers of larvae reported were 12.3 [19,20] and 24.5 larvae [54,87,89], respectively.

### 4.5. Treatment

There is no specific treatment for ophthalmomyiasis. Nevertheless, in post-larval removal, it is advisable to provide anti-inflammatory drugs, antibiotics, and ivermectin, as they have been successfully employed in documented cases [47,65,87,134] for the past two decades.

External ophthalmomyiasis requires immediate removal of the larvae under local anesthesia and the application of antibiotics to prevent bacterial infection. Anesthetic drops can paralyze the maggots and facilitate their removal [135]. Even though patients may experience immediate relief, it is essential to have a follow-up examination within 24 to 48 h to ensure that any remaining larvae in the conjunctival sac fundus are treated. If any larvae are left behind, the initial treatment may not be effective [1]. Despite the painful and alarming symptoms, external ophthalmomyiasis can be successfully cured [136].

Our review found that despite the high number of larvae in cases of external ophthalmomyiasis caused by *O. ovis*, *C. bezziana*, or *L. sericata*, all reported cases were completely cured. Anane and Hssine [1] reported 11 cases of complete cures in 2020, highlighting the importance of prompt treatment and complete removal of the larvae to prevent complications. Therefore, seeking early treatment for external ophthalmomyiasis cases is imperative to prevent further complications.

Based on the evidence reviewed, the initial treatment step is recommended to consist of larvae being immediately immobilized with local anesthetic eye drops and extracted. The second step is to apply antibiotics and topical anti-inflammatories to alleviate symptoms and prevent secondary infection. This recommended treatment sequence aligns with previous publications by Thakur et al. [3] and Abdellatif [18]. To avoid potential complications or recurrence, scheduling a follow-up examination is recommended [1].

To effectively treat a *D. hominis* infestation, immediate surgical removal of the larvae is strongly recommended to minimize inflammation and the possibility of requiring more extensive surgery. It is essential to thoroughly examine patients for additional skin infestations [17,55,111,120]. Symptoms subside once the larvae are removed, and a favorable prognosis is expected [130].

### 4.6. Travelers

Myiasis is a common dermatologic disease among travelers, and ophthalmomyiasis accounts for 5% of all cases of myiasis [98,99]. External ophthalmomyiasis related to tourism accounts for 8.9% of cases. Therefore, if a tourist returns from an endemic area such as India [37], Jordan [88], Libya [18], Turkey [26], Iran [75], Tunisia [1], Spain [137], Italy [98], and French Guiana [17] with symptoms such as red eye, unilateral conjunctivitis, or foreign body sensation, physicians should consider external ophthalmomyiasis as a possible cause and conduct a thorough examination [59]. Physicians in non-endemic regions unfamiliar with this infestation may find this challenging, especially when other organs besides the skin are affected [10]. Therefore, it is crucial to remain vigilant and not overlook this possibility.

### 4.7. Military Deployments

With the increasing frequency and duration of military deployments worldwide, civilian and military medical professionals must know the potential hazards of exposure to various conditions. Some conditions, such as ophthalmomyiasis, may not be commonly diagnosed by medical personnel [58,106]. To prevent insect-borne disease transmission, personnel deployed to war zones and areas with insect vectors should equip themselves with protective eyewear [138].

### 4.8. Global Warming

In the last two decades, an increased prevalence of oestrosis in humans and sheep have been observed in endemic areas, as well as a broader geographic expansion into areas not previously considered endemic. However, the appearance of new cases is attributed to the effect of global warming creating a suitable habitat for the survival of *O. ovis* and not to the increase in notifications as shown by Ahaduzzaman [116], Weaned [108], Basmaciyan et al. [117], Sucilathangam et al. [38], Zhang [49] and Taylor [118].

Cases linked to the effects of global warming have been reported in Germany and France, specifically concerning *O. ovis* [108] and *Oestrus* sp. [117], causing external ophthalmomyiasis. Surprisingly, despite these regions’ typically cold and wet climates, significant warming during spring and summer has allowed for the establishment of these flies. Furthermore, temperature changes between 1961 and 2011 in Burgundy [139], France, showed a more accelerated warming trend than the global average, leading to the implantation of Diptera of the genus *Oestrus* sp. during the vegetative season. This warming is more pronounced in diurnal temperatures and has impacted the establishment of these Diptera in regions previously considered cold and humid [140]. Another case associated with global warming is a 30-year-old man in Shandong [49], China, who experienced autochthonous external ophthalmomyiasis despite the city being cold and dry for most of the year. Therefore, global warming is significantly impacting the establishment and spread of Diptera, which is a cause for concern. Action must be taken to address this issue before it is too late.

Global warming has affected the behavior of *O. ovis*, causing them to change their egg-laying habits. Due to the human facial structure, these flies confuse our pupils with cattle nostrils, leading to increased ophthalmomyiasis cases. It is worth noting that even individuals who have not had contact with animals or visited areas where the disease is prevalent have been affected. These occurrences have been documented in [45,106].

The attraction of *O. ovis* to humans is underestimated [141]. *O. ovis* is an insect that infests urban areas due to the zooprophylaxis phenomenon. Humans become the primary target for these flies when there is a shortage of other hosts, thereby increasing the risk of infection [142]. Various weather conditions, such as air temperature, humidity, light intensity, altitude, latitude, and wind speed, influence this type of fly. Attacks usually occur when the air temperature is above 20 °C, with the highest frequency between 25 °C and 28 °C. The most frequent attacks occur at an air temperature of 26 °C and under calm to moderate wind conditions. Furthermore, the relative humidity range also affects the fly, with the highest peaks between 65% and 85%. Climate change has been identified as the main factor influencing the behavior of *O. ovis* in several regions, increasing temperature and the fly’s infestation rates [104,143].

On the other hand, reported cases of *D. hominis* infestation are preceded by a history of travel or residence in Latin America, areas to which *D. hominis* is native [17,55,120,130]. Additionally, it has been suggested that possible global warming in North America could result in a migration of this species to areas previously considered unsuitable habitats to allow it to thrive [112].

### 4.9. Prevention

Ophthalmomyiasis is not a minor disease and poses a significant public health concern, especially in developing countries such as India [5,27,29,37,38,62,63,64,65,66,67,79], Jordan [42,88], and Turkey [25,26,28,68,69,72,89,90,91,92,144]. However, the larvae responsible for this disease are often discarded without proper examination [10], leading to inadequate case recording. Access to Dipteran classification experts can be challenging in such regions. It is important for ophthalmologists to include ophthalmomyiasis in their differential diagnosis of conjunctivitis [90] and for physicians to remember that this condition can occur not only in tropical or subtropical areas but also in other non-endemic regions [25,70,108,145] with more temperate climates. Diagnosis of ophthalmomyiasis heavily relies on expertise [26]. Most external ophthalmomyiasis lesions occur in exposed areas, so using mosquito repellent [145] and nets for windows and ventilation ducts [19] can be the best preventive measures.

Preventing the spread of disease can be achieved by avoiding exposure to adult flies and using insecticide spray to kill them. Raising awareness is crucial in combatting this preventable endemic disease in rural areas [37,65,88,92]. Travelers must be informed of the necessary preventive measures based on their destination to avoid myiasis [48,55,57,59,99]. It is advisable to avoid endemic countries where farm animals are left unattended to reduce the risk of contracting myiasis. If any symptoms of myiasis are experienced, consulting a physician is recommended. Additionally, physicians should consider ophthalmomyiasis as a possible cause of conjunctivitis in tourists visiting endemic areas or in individuals working with farm animals, such as farmers, ranchers, and veterinarians [90].

On the other hand, it is essential to avoid myiasis infestation in unconscious patients who are hospitalized. To avoid this, institutions must actively combat flies in clinics or hospitals. Windows should be closed with mosquito nets to prevent the entry of insects. Staff working in intensive care units where there is a significant number of unconscious patients must daily check body cavities such as the eyes, nose, mouth, and ears, which present a risk of larvae being deposited in unconscious patients, and carry out their care and cleaning properly [22,23].

Animal treatment should be performed routinely in sheep and goat flocks in regions with a high disease burden with the aim of improving animal productivity and minimizing the potential zoonotic risk to humans. Additionally, appropriate preventive measures against *O. ovis* infestation must be implemented. As oestrosis is more common in adult animals, the development of vaccines and immunization can prevent the disease. Furthermore, reports on the prevalence of oestrosis in sheep and goats are not yet available in many regions; therefore, epidemiological surveillance is needed to estimate the disease burden and control it [116].

### 4.10. Limitations

It is likely that more cases of ophthalmomyiasis are caused by *O. ovis* than what is currently documented. Many cases lack crucial details, such as the time of diagnosis or the condition’s cause. Underreporting is a leading factor in this issue, as patients with ophthalmomyiasis caused by *O. ovis* often do not seek medical attention. Additionally, the larvae responsible for the condition die within the first ten days of infestation in humans, which could lead to misdiagnosis and further underestimation of cases.

## 5. Conclusions

It can be challenging to determine the exact number of individuals impacted by external ophthalmomyiasis because cases caused by *O. ovis* usually have mild symptoms that are easily resolved by removing the larvae. As a result, patients typically recover completely and do not experience any lasting effects. This means that some cases may not be recorded or reported, leading to a lower estimated global number of affected individuals.

External ophthalmomyiasis is a commonly underestimated condition, especially in dry rural areas. It is often misdiagnosed as viral, bacterial, or allergic conjunctivitis, which can be detrimental to the patient’s health. Therefore, physicians must be aware of its prevalence and symptoms. Early recognition and a greater medical understanding are necessary to prevent the development of internal ophthalmomyiasis, a severe condition that can significantly affect the patient’s visual prognosis.

Individuals residing in regions with high fly larvae infestation rates are at significant risk of contracting ocular infections. Physicians must consider the possibility of ophthalmomyiasis when diagnosing anterior segment inflammation. Travelers must know that ophthalmomyiasis can be averted by taking appropriate preventive measures based on their destination. Medical professionals must also caution travelers visiting endemic regions about the risk of myiasis if they fail to undertake preventive measures.

Moreover, it is crucial to identify autochthonous cases of external ophthalmomyiasis in non-endemic areas to evaluate the impact of climate change on establishing Diptera, specifically the *Oestrus* genus.

## Figures and Tables

**Table 1 diseases-11-00180-t001:** Geographical distribution and agents responsible for external ophthalmomyiasis from 2000 to 2022.

Country	Number of Reported Cases	Percentage of Cases	Causative Agents
India	62	19.9	*Oestrus ovis*, *Musca domestica*, *Chrysomya bezziana*
Jordan	50	16.0	*Oestrus ovis*
Turkey	45	14.4	*Oestrus ovis*, *Chrysomya bezziana*
Iran	27	8.7	*Oestrus ovis*, *Lucilia sericata*
Libya	22	7.1	*Oestrus ovis*
Tunisia	12	3.8	*Oestrus ovis*
French Guiana	10	3.2	*Dermatobia hominis*
Italy	10	3.2	*Oestrus ovis*
Spain	10	3.2	*Oestrus ovis*, *Dermatobia hominis*
USA	8	2.6	*Dermatobia hominis*, *Oestrus ovis*, *Phaenicia lucilia*
Germany	6	1.9	*Oestrus ovis*
France	6	1.9	*Oestrus ovis*
Peru	5	1.6	*Oestrus ovis*
Israel	4	1.3	*Oestrus ovis*, *Sarcophaga argyrostoma*
Brazil	4	1.3	*Dermatobia hominis*, *Cochliomyia macellaria*
Oman	2	0.6	*Oestrus ovis*
Republic of Croatia	2	0.6	*Oestrus ovis*
Saudi Arabia	2	0.6	*Oestrus ovis*
Mexico	2	0.6	*Oestrus ovis*
Afghanistan	2	0.6	*Oestrus ovis*
Pakistan	2	0.6	*Oestrus ovis*
Nepal	2	0.6	*Oestrus ovis*
Czech Republic, Bulgaria, South Africa, Australia, China, Jamaica, United Kingdom, Iraq, Bolivia, Belgium, Barbados, and Morocco.	1 *	3.6	*Oestrus ovis*
Serbia, South Korea	1 *	0.6	*Lucilia sericata*
Canada	1 *	0.3	*Dermatobia hominis*
Indonesia	1 *	0.3	*Chrysomya bezziana*
Japan	1 *	0.3	*Boettcherisca peregrine*
Honduras	1 *	0.3	*Not identified*
Total	312	100.00	

* All of these countries each reported one case of ophthalmomyiasis during the period 2000 to 2022.

**Table 2 diseases-11-00180-t002:** Treatment used in cases of external ophthalmomyiasis caused by *O. ovis*, *D. hominis*, *L. sericata* and *C. bezziana* from 2000 to 2022.

*Oestrus ovis*	n (%)	*Dermatobia hominis*	n (%)	*Lucilia sericata*	n (%)	*Chrysomya bezziana*	n (%)
Manual extraction + local antibiotic	109 (52.8)	Surgical removal	10 (58.8)	Manual extraction + local antibiotic	2 (66.7)	Manual extraction + local antibiotic	2 (66.7)
Manual extraction + local antibiotic + local steroid	86 (41.3)	Surgical removal + antibiotic	4 (23.5)	Manual extraction + local antibiotic + steroid	1 (33.3)	Manual extraction + local antibiotic + ivermectin	1 (33.3)
Manual extraction	6 (2.9)	Surgical removal + ivermectin	2 (11.8)				
Manual extraction + analgesic + local antibiotic	4 (1.9)	Surgical removal + antibiotic + steroid	1 (5.9)				
Manual extraction + ivermectin	1 (0.5)						
Manual extraction + ivermectin + local antibiotic	1 (0.5)						
Manual extraction + tobacco infusion	1 (0.5)						

n: number of cases treated; %: Percentage of cases treated.

## Data Availability

Not applicable.

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
