# Peer review of "Ophthalmomyiasis Externa and Importance of Risk Factors, Clinical Manifestations, and Diagnosis: Review of the Medical Literature"

_diseases, 2023, doi:10.3390/diseases11040180_

Round 1
Reviewer 1 Report
Comments and Suggestions for Authors
Dear Authors
The study of myiasis holds significant medical importance due to the risks it poses to human and animal health. Early diagnosis and appropriate treatment are essential to prevent complications. Therefore, studying myiasis is crucial for effective prevention, diagnosis, and treatment of this condition, contributing to public and veterinary health.
I would like to stress that I support the potential publication of this paper due to its scientific interest. It is clear that there has been prior work in this field, and the authors summarize the significant findings from others and how the current study is unique and adds to what has been described.
A minor change in the title is suggested in the manuscript. The title mentions the emphasis on global warming and diagnosis, but in the manuscript, equal importance is given to other parts of the disease. The first reading of the title suggests that it will essentially focus on these parts (global warming and diagnosis). I suggest introducing other terms in the title, such as risk factors and clinical manifestations.
I also don't understand why "Mexico" was included in the keywords when the review is made about the entire world. This word should be removed because anyone who reads it will believe that the research is only about Mexican articles (for example, Table 1). I understand the Authors are from Mexico, but the work has international soundness.
Tables and references can be better formatted according to the journal.
The results of the study justify the discussion interpretations and conclusions.
Author Response
Reply to first reviewer
Thank you very much for your valuable comments.
|
1. A minor change in the title is suggested in the manuscript. The title mentions the emphasis on global warming and diagnosis, but in the manuscript, equal importance is given to other parts of the disease. The first reading of the title suggests that it will essentially focus on these parts (global warming and diagnosis). I suggest introducing other terms in the title, such as risk factors and clinical manifestations. |
Thanks for the suggestions; changes were made to the title of the manuscript. |
|
2. I also do not understand why "Mexico" was included in the keywords when the review is made about the entire world. This word should be removed because anyone who reads it will believe that the research is only about Mexican articles (for example, Table 1). I understand the Authors are from Mexico, but the work has international soundness. |
The word Mexico was eliminated. |
|
3. Tables and references can be better formatted according to the journal. |
Improved formatting of tables and references. |
Sincerely
Professor Hugo Martínez-Rojano

Reviewer 2 Report
Comments and Suggestions for Authors
The submitted manuscript presents a review whose main aim is to summarize available knowledge and recent findings on case reports and discussion on the clinical presentation of human external ophthalmomyiasis.
The review presents an updated state-of-the-art on the subject with manuscript sections including an introductory section in which it is indicated that in addition to performing a comprehensive literature review providing information on the various risk factors, clinical course, diagnosis, parasitological and environmental characteristics, and treatment options for this condition, preventive measures as well as the effect of global warming are discussed, based on a case series reported between January 2000 and December 2022.
Thus, this review discusses available information on most of the reported cases of ophthalmomyiasis around the world, and a timely and updated review on the increasing number of case reports on external ophthalmomyiasis in people, mainly as a result of fly expansion associated with climate change.
Major headings of the manuscript describe a review of the literature published, focusing on results containing Epidemiological information, Clinical features, Diagnosis, Treatment, Complications, and a general discussion section describing and comparing the risk factors associated with the presentation of ophthalmomyiasis cases; the etiologic agents involved in such reported cases, the clinical manifestations observed in affected people, the diagnostic approaches used, treatment, presentation of cases in travelers and military personnel, and a section about the effect of global warming on the presentation of clinical cases of external ophthalmomyiasis, as well as the preventive measures indicated to avoid exposure to the adult flies. The review is supported by tables describing the geographical distribution and agents responsible for external ophthalmomyiasis, as well as the treatment used in cases of external ophthalmomyiasis caused by larvae of Oestrus ovis, Dermatobia hominis, Lucilia sericata , and Chrysomyia bezziana from 2000 to 2022.
Authors revised 160 references that have been published on the matter which cover most if not all of the literature published on reported cases of ophthalmomyiasis, particularly during the period comprehended from January 1, 2000, to December 31, 2022, which could be useful to update the current knowledge about ophthalmomyiasis.
Authors searched for articles on the subject in electronic databases such as Latin American and Caribbean Literature in Health Sciences, Scientific Electronic Library Online, PubMed, EBSCO-host, and Google Scholar, as well as in individual journals, by including articles in English, German, Turkish, French, and Italian, on external ophthalmomyiasis, external ocular myiasis, and conjunctival myiasis.
Overall, the review is well written, summarizes the literature published on the ophthalmomyiasis topic, discusses it critically, identifies major methodological problems and points out existent clinical research gaps.
The topic presented in the review falls within the aims and scope of the Diseases journal, and it is the reviewer´s opinion that the article is suitable for publication.
Corrections that authors need to take care of are primarily related to the References:
Authors are asked to follow the instructions to authors with regard to how the references must be numbered in the text (including table captions and figure legends), in order of appearance, and listed individually at the end of the manuscript. Up until citation 61 (line 238), all references are numbered and listed OK. However it is from lines 239 and 240 that citations begin to get scrambled, see for example line 239 [2,27,29,37,69,70,78,79,142,144]; 240 [28,43,81-87,157], [20,40,62], and [18,63]. Authors should carefully check all references and renumber them from here on throughout the manuscript.
Once the citations are re-ordered in the manuscript and correctly numbered in the references section, authors should also check that all references are correctly described as follows:
Journal Articles:
1. Author 1, A.B.; Author 2, C.D. Title of the article. Abbreviated Journal Name Year, Volume, page range.
Comments on the Quality of English Language
To my opinion, the English language is fine, and minor changes are required.
Author Response
Responses to the second reviewer
Thank you very much for your valuable comments.
|
1. Once the citations are re-ordered in the manuscript and correctly numbered in the references section, authors should also check that all references are correctly described as follows: Journal Articles: 1. Author 1, A.B.; Author 2, C.D. Title of the article. Abbreviated Journal Name Year, Volume, and page range. |
Citations were rearranged in the manuscript and numbered correctly in the references section. The correct description of the references was verified. |
Sincerely
Professor Hugo Martínez-Rojano

Reviewer 3 Report
Comments and Suggestions for Authors
Title: Ophthalmomyiasis externa and importance of global warming and diagnosis: review of the medical literature
Authors: Hugo Martinez-Rojano et al.
The proposed corrections are following:
Line 31 to 33, it is not clear whether the species Dermatobia hominis belongs to the family Oestridae or Calliphoridae. I suggest that the name of the species D. hominis be given immediately after the name of the species Oestrus ovis and only then followed by the name of the order and family (Diptera: Oestridae). Do the same for the species C. bezziana, it is necessary to state the name of this species after the species name of L. sericata, and only then follow the name of the order and family (Diptera: Calliphoridae).
Line 33, L. sericata, the author's name is missing after the species name, it is necessary to write Meigen
Line 57 to 59, for obligate myiasis, the name of the genus Wohlfahrtia from the Sarcophagidae family should be mentioned. Accordingly, reference should be made to Hall M.J.R. and Smith K.G.V. 1993. Diptera causing myiasis in man. In: Medical insects and arachnids. Lane R.P. and Crosskey R.W. (Eds.). Chapman & Hall, London, 723 pp.
Line 85 genera name Curterebra should be written in italic, also Gasterophilus and Hypoderma in line 86 and 87.
Line 86 in name of family Gasterophilidae missing letter “e”
Line 121 instead of population write inhabitants
Line 329 genera name Curterebra should be written in italic
Line 386 the reference for data on the number of diptera is missing, I suggest the reference: Durbešić P. 1988. Acquaintance and research of terrestrial arthropods. Mala ekološka ekološka biblioteka, knjiga 4, Hrvatsko ekološko društvo, Zagreb, 77pp.
Author Response
Responses to third reviewer
Thank you very much for your valuable comments.
|
1. Line 31 to 33, it is not clear whether the species Dermatobia hominis belongs to the family Oestridae or Calliphoridae. I suggest that the name of the species D. hominis be given immediately after the name of the species Oestrus ovis and only then followed by the name of the order and family (Diptera: Oestridae). Do the same for the species C. bezziana, it is necessary to state the name of this species after the species name of L. sericata, and only then follow the name of the order and family (Diptera: Calliphoridae). |
The proposed suggestions were accepted. |
|
2. Line 33, L. sericata, the author's name is missing after the species name, it is necessary to write Meigen |
The proposed suggestion was made |
|
3. Line 57 to 59, for obligate myiasis, the name of the genus Wohlfahrtia from the Sarcophagidae family should be mentioned. Accordingly, reference should be made to Hall M.J.R. and Smith K.G.V. 1993. Diptera causing myiasis in man. In: Medical insects and arachnids. Lane R.P. and Crosskey R.W. (Eds.). Chapman & Hall, London, 723 pp. |
The proposed suggestion was made |
|
4. Line 85 genera name Curterebra should be written in italic, also Gasterophilus and Hypoderma in line 86 and 87. |
The proposed suggestion was made |
|
5. Line 86 in name of family Gasterophilidae missing letter “e” |
The proposed correction was made |
|
6. Line 121 instead of population write inhabitants |
The proposed correction was made |
|
7. Line 329 genera name Curterebra should be written in italic |
The proposed suggestion was made |
|
8. Line 386 the reference for data on the number of diptera is missing, I suggest the reference: Durbešić P. 1988. Acquaintance and research of terrestrial arthropods. Mala ekološka biblioteka, knjiga 4, Hrvatsko ekološko društvo, Zagreb, 77pp. |
The proposed suggestion was made |
Sincerely
Professor Hugo Martínez-Rojano

Reviewer 4 Report
Comments and Suggestions for Authors
There are many recent reviews on Oestrus ovis infection in animals (Gracia et al., 2019 mostly in sheep or Ahaduzzaman -in sheep and goats with a meta-analysis). The veterinary reviews often have review of human cases as well. The medical cases, which are limited in scope and numbers, often add a mini review of other human cases. Thus, literature is replete with reviews. The present one includes global warming as a factor of prevalence increase. I did not see anything on the increase of prevalence due to global warming; there is one of decrease in sheep (Jacquiet et al 2002). The records of O. ovis under very different climates shows the plasticity of this insect regarding temperature. This should be deleted from the title.
The text is lengthy and there are many repetitions among the different chapters, mostly on symptoms. It could be reduced by half from 24 to 12 pages by concentrating on the risk factors and possibly on an accurate semiology to improve diagnosis. The risk factors are “analysed” one by one and in terms of percentages. From a quick look at the supplementary file reporting the different cases, I think that risk factors should be taken all together using a general linear model or multidimensional analysis. The factors are not independent: shepherds are mostly men for example. Most of the data are on O. ovis and the meta-analysis should concentrate on this parasite.
There is one inconsistency in introduction: “The most common type of myiasis among humans is caused by D. hominis” vs “O. ovis, … has a cosmopolitan distribution and has been reported to be the most common causative agent of ophthalmomyiasis externa”. In table 1explain what is number and percentage.
Comments on the Quality of English LanguageThe English is quite understandable but could be revised by a native speaker.
Author Response
Responses to the fourth reviewer
Thank you very much for your valuable comments.
|
1. The present one includes global warming as a factor of prevalence increase. I did not see anything on the increase of prevalence due to global warming; there is one of decrease in sheep (Jacquiet et al 2002). The records of O. ovis under very different climates shows the plasticity of this insect regarding temperature. This should be deleted from the title. |
The term global warming was removed from the title. |
|
2. The text is lengthy and there are many repetitions among the different chapters, mostly on symptoms. It could be reduced by half from 24 to 12 pages by concentrating on the risk factors and possibly on an accurate semiology to improve diagnosis. |
The text of the manuscript is 13 pages; the rest corresponds to the references. The first section presents the results of the review that was carried out; the second section presents the discussion, where the results described in the first section are contrasted. |
|
3. The risk factors are “analysed” one by one and in terms of percentages. From a quick look at the supplementary file reporting the different cases, I think that risk factors should be taken all together using a general linear model or multidimensional analysis. The factors are not independent: shepherds are mostly men for example. Most of the data are on O. ovis and the meta-analysis should concentrate on this parasite. |
Thank you very much for your proposal |
|
4. There is one inconsistency in introduction: “The most common type of myiasis among humans is caused by D. hominis” vs “O. ovis, … has a cosmopolitan distribution and has been reported to be the most common causative agent of ophthalmomyiasis externa”. In table 1explain what is number and percentage. |
In the case of myiasis, the most common causative agent is D. hominis. Ophthalmomyiasis corresponds to approximately 5% of myiasis cases and the most common causative agent is O. ovis. |
Sincerely
Professor Hugo Martínez-Rojano

Round 2
Reviewer 4 Report
Comments and Suggestions for Authors
The authors have provided an improved version of their paper. The table with the,percentages include reports with only one case of O. ovis (China for example) and I wonder about the details on these unique cases, they could be regrouped.
Comments on the Quality of English LanguageMinor. When you cite authors in the manuscript you should not write the initials of christian names.
Author Response
Cover letter
Diseases-2691671
Thank you very much for all the valuable comments.
Response to each of the observations made by the reviewer
The authors have provided an improved version of their paper. The table with the, percentages include reports with only one case of O. ovis (China for example) and I wonder about the details on these unique cases, they could be regrouped.
Reply: Countries where a case has been reported are grouped according to the causal agent
Minor. When you cite authors in the manuscript, you should not write the initials of Christian names.
Reply: The initials of the name were removed in the case where an author is cited.
Professor Hugo Martínez-Rojano
Higher School of Medicine
Institute of Epidemiological Diagnosis and Reference
